# Characteristics of the Molten Pool Temperature Field and Its Influence on the Preparation of a Composite Coating on a Ti6Al4V Alloy in the Micro-Arc Oxidation Process

**DOI:** 10.3390/ma13020464

**Published:** 2020-01-18

**Authors:** Xinyi Li, Chaofang Dong, Qing Zhao, Fasong Cheng

**Affiliations:** 1Corrosion and Protection Center, University of Science and Technology Beijing, Beijing 100083, China; 15976659090@163.com; 2National Defense Key Discipline Laboratory of Light Alloy Processing Science and Technology, Nanchang Hangkong University, Nanchang 330063, China; 3AECC Guizhou Liyang Aviation Power Co. Ltd., Guizhou 550007, China; bkdcf1976@126.com

**Keywords:** micro-arc oxidation, temperature field, molten pool, secondary phase particles, composite coating

## Abstract

In this study, the phase transition of secondary phase particles in a composite coating is used to estimate the temperature field of the molten pool on a Ti6Al4V alloy in the micro-arc oxidation (MAO) process. The behavior of the sparks and the molten pool during the MAO process was observed in real-time by a long-distance microscope. The microstructures and compositions of the composite coatings were studied by scanning electron microscope (SEM), energy dispersive spectroscopy (EDS), and X-ray diffraction (XRD). The results revealed that, for the temperature field distribution range of the molten pool in the active period, the lower limit is 2123 K and the upper limit is not lower than 3683 K. The reason for the multiphase coexistence is that the high-temperature phase is retained by the rapid cooling effect of the electrolyte, and the low-temperature phase is formed due to secondary phase transformation during the long active time of the molten pool temperature field. The strengthening mechanism of the composite coating prepared by adding the secondary phase particles is elemental doping rather than particle enhancement. The secondary phase particles are able to enter the composite coating by adhering to the surface during the cooling process. The secondary phase particles will then be wrapped into the coating in the next active period.

## 1. Introduction

Micro-arc oxidation (MAO) is an emerging technique used to prepare oxidized ceramic coatings in situ on valve metals such as titanium, magnesium, aluminum, and their alloys. Through applying a high potential, which is much larger than that of the Faraday region of conventional anodization, an oxidized ceramic coating is obtained by the effect of thermochemical, electrochemical, and plasma chemistry in a specific electrolyte [1,2,3,4,5]. Due to the loss of the dielectric stability of the sample in the area of low conductivity, when the given voltage is higher than the critical value, a discrete spark discharge occurs on the surface of the sample. Molten materials are ejected from the channels at higher discharge temperatures and pressures, and they rapidly cool due to their direct contact with the electrolyte, resulting in an overall increase in coating thickness [6,7,8]. The MAO technique is commonly used to increase the wear resistance on the surface of aluminum, magnesium, and titanium alloys [9,10,11]. Usually, MAO coatings are multiphase, even without any additional external particles. They include amorphous and various crystalline phases (rutile and anatase). Doping the secondary phase particles in the MAO coating can further improve the tribological properties and even give the coating new functions, which has become a popular research direction [12,13,14]. Secondary phase particles such as Al_2_O_3_ [15], SiC [16], Cr_2_O_3_ [17], BN [18], and ZrO_2_ [13] can be added. The way of entering the MAO coating, the phase transition, and the existing state of secondary phase particles in the molten pool are closely related to the temperature field characteristics of the micro-arc oxidation pool.

Hussein et al. [19] used optical emission spectroscopy to study the electron beam temperature of MAO of a 1100 aluminum alloy, and found that the electron beam temperature is between 4000 K and 7000 K in the unipolar current mode, and is between 4000 and 5500 K in the bipolar current mode. Klapkiv et al. [20] used emission spectra to evaluate the micro-arc oxidation plasma temperature of pure aluminum between 6899 and 7700 K. Hussein et al. [21] measured the micro-arc oxidation plasma temperature of an AJ62 magnesium alloy and showed that it is between 3300 and 7000 K by optical emission spectroscopy. However, this non-contact method of temperature measurement is not precise and has not been effectively confirmed.

In this paper, the phase transition temperature of the secondary phase particles is used to determine the temperature of the molten pool, although the actual temperature is necessarily higher than the phase transition temperature. Factors such as the continuous active time of the molten pool, the geometry of the molten pool, and the density of the molten pool will affect the existing state of the secondary phase particles. The aim of this work is to study the characteristics of the molten pool and its impact on the coating preparation in the micro-arc oxidation process, which is crucial for the design of the composite MAO coating.

## 2. Materials and Methods

The experimental material was a Ti6Al4V alloy, and its main chemical composition is shown in Table 1. The samples with dimensions of 20 mm × 20 mm × 1.5 mm were polished with sandpaper from 240# to 1200# and then ultrasonically cleaned with absolute ethanol for 30 min. The degreased samples were stored in alcohol for the subsequent MAO process.

In order to uniformly distribute the secondary phase particles in the electrolyte, it is necessary to degrease the secondary phase particles and disperse the surfactant. The characteristics of the secondary phase particles and the electrolyte composition are shown in Table 2 and Table 3, respectively. The secondary phase particles were mixed with NaOH to conduct degreasing, according to the ratio of 100:1 (wt.%). Deionized water was added to the mixture and stirred for 20 min. Then, the mixture was filtered and an anionic surfactant was added to disperse the particles. After ultrasonic dispersion for 20 min, it was filtered with deionized water and dried for 2 h at 100 °C.

A DC pulse power supply (WD-20, GUIGE, Harbin, China) was used for MAO treatment on the Ti6Al4V alloy. The current density was set to 10 A/dm^2^. The pulse frequency was 500 Hz and the duty cycle was 50%. The MAO time was 65 min. The choice of MAO time was based on the aim to study the duration and distribution of the spark and molten pool.

The QUESTAR QM100 long-distance microscope (QUESTAR, New Hope, PA, USA) was used to observe the spark and molten pool in real-time. The lens-to-target distance was 15–35 cm and a clear image could be obtained by adjusting the viewing distance. In order to collect and record the morphology of the spark and the molten pool during the MAO process, the Microshot MDX-4T image collector (Microshot, Shenzhen, China) with a resolution of 800 × 600 and a single-pixel size of 1.4 × 1.4 μm was connected to a long-distance microscope. The frame rate was 28 frames per second.

The compositions of the obtained coatings were analyzed by X-ray Diffraction (XRD) using a Bruker D8-Adance instrument (Bruker, Billerica, MA, USA). The surface morphologies of the coatings were observed by scanning electron microscopy (Quanta 200, FEI, Hillsboro, OR, USA). The local components were characterized by energy-dispersive spectroscopy (EDS) (INCA-250, Oxford instruments, Oxford, UK).

## 3. Results and Discussion

### 3.1. Phase Transition of Secondary Phase Particles under the Temperature Field Effect of the Molten Pool

Figure 1 shows the XRD patterns of the composite coatings prepared by adding γ-Al_2_O_3_ and κ-Al_2_O_3_. The α-Al_2_O_3_ could be found in two kinds of composite coatings, which proves that the two phase transitions occurred at 1303 and 1288 K, according Table 2. The amorphous phase was formed during the MAO process under the effect of discharge with high energy. The results show that under the effect of the molten pool temperature field, both γ-Al_2_O_3_ and κ-Al_2_O_3_ were transformed into α-Al_2_O_3_, and the molten pool temperature should be higher than 1303 K.

Figure 2 shows the XRD patterns of the composite coating prepared by adding m-ZrO_2_. As shown in Figure 2, the as-prepared coating included m-ZrO_2_, t-ZrO_2_, and c-ZrO_2_, indicating that partial m-ZrO_2_ was transformed into t-ZrO_2_ and c-ZrO_2_ under the effect of the molten pool temperature field. However, partial m-ZrO_2_ still existed. This result could have been achieved through two possible scenarios. The first possibility is that the temperature of the partial region was higher than 2643 K, meaning that m-ZrO_2_ was transformed into c-ZrO_2_. The temperature of the partial region was higher than 1473 K, but lower than 2643 K, meaning that m-ZrO_2_ was transformed into t-ZrO_2_. No phase transition occurs in areas where the temperature is less than 1473 K. The second possibility is that under the high temperature of the molten pool, all m-ZrO_2_ was transformed into c-ZrO_2_. However, due to the cooling effect of the electrolyte, phase transformation occurred again as the temperature decreased. Part of c-ZrO_2_ was transformed into t-ZrO_2_ and m-ZrO_2_.

Figure 3 shows the XRD pattern of the composite coating prepared by adding β-SiC. As shown in Figure 3, the as-prepared composition coating included β-SiC, α-SiC, and TiC. According to Table 2, β-SiC will be transformed into α-SiC at 2373 K, and will decompose into Si and C at a temperature higher than 3143 K. The existence of TiC illustrates that β-SiC was decomposed into Si and C, which proves that the temperature of the molten pool can reach 3143 K.

The matrix of the titanium alloy MAO composite coating is titanium oxide. In Figure 1, Figure 2 and Figure 3, Rutile-TiO_2_ and Anatase-TiO_2_ are both found. Rutile-TiO_2_ with a higher melting point dissolved at 2123 K. The MAO coating was formed by the solidification of Rutile-TiO_2_, indicating that the temperature of the molten pool must be higher than 2123 K. In our previous report [23], the temperature of the molten pool was estimated to be higher than 3683K. In the later stage of growth of the MAO coating, the size of the sparks and the molten pool were at the sub-millimeter level [24]. In summary, for the temperature field distribution range of the molten pool in the active period, the lower limit is 2123 K, and the upper limit is not less than 3683 K.

### 3.2. Duration and Distribution of the Spark and Molten Pool

When the spark breaks through the surface of the micro-arc oxide coating, a corresponding molten pool is generated [25]. With the progress of micro-arc oxidation, the size of the spark and the molten pool increased uniformly, but the size of the molten pool was slightly larger than the size of the spark. Three typical moments of micro-arc oxidation were chosen for discussion. At 5 min, the voltage increased rapidly; at 20 min, the voltage turned; and at 65 min, the voltage was stable and slowly increased.

Figure 4a shows an optical image of the Ti6Al4V substrate. Figure 4b–d illustrate the anodizing film with different colors on the surface of Ti6Al4V, due to light interference with different thicknesses. Figure 4e–g show the images of the spark at 5, 20, and 65 min, respectively.

Figure 5 shows an optical photograph of the spark and molten pool taken with a long-distance microscope. The duration of the spark was obtained by real-time observation. The density of the spark was calculated by statistical methods. A molten pool was formed at the position after the spark was extinguished. The sizes of the spark and the molten pool were measured. The results are summarized in Table 4.

In the early stage of the MAO process, the density of the sparks is large, and the size of the spark and the molten pool is small. The active time of the sparks and the molten pool is short, which is not conducive to larger particles recombining into the MAO coating and undergoing sufficient phase transformation. In the later stage, the size of the spark and the molten pool is large enough, and the active time is longer, which is conducive to the secondary phase particles with a large size entering the MAO coating [17]. Therefore, micron and submicron-sized particles are more suitable for preparing micro-arc oxidation composite coatings.

Figure 6 shows the surface images of the MAO coating before and after the spark discharge at 50 min. Figure 6b–d show the images at 0.14 s, 0.84 s, and 0.91 s after the time shown in Figure 6a, respectively. The blue circle area “m” and area “n” are the weak regions of the MAO coating, where coating morphology changes occur during the spark discharge. During the process of micro-arc oxidation, the sparks correspond to the generation of molten pools [15]. In order to further study the mechanism, the surface microscopic morphology of the coating was shown in Figure 7. The large spark consists of clusters of many small sparks. The reason for this may be that the conductivity of titanium dioxide rises with an increasing temperature, and the electrical conductivity is larger near the large spark, prone to coating breakdown, and generates new small sparks. The larger spark lasts a longer time due to the smaller sparks being generated continuously. It can be simply assumed that the sizes of the spark and the molten pool are similar, and the active time is also similar. Therefore, the space and time in which they occupy secondary phase particles with micron and submicron particle diameters results in the effects of a higher temperature and longer time, leading to sufficient phase transition and dissolution. In the composite coating prepared with secondary phase particles with micron and sub-micron particle diameters, it is easy to obtain a higher concentration of the secondary phase particle element composition in the cross-sectional SEM image (Figure 8), but it is difficult to find the secondary phase particles. The secondary phase particles have been completely dissolved under the temperature field of the molten pool.

### 3.3. The Way That the Secondary Phase Particles Enter the MAO Coating

m-ZrO_2_ and γ-Al_2_O_3_ particles were used to prepare the composite coatings. In the first step, the sample was put into the electrolyte containing m-ZrO_2_ particles for MAO 30 min. In the second step, the above sample was put into the electrolyte containing γ-Al_2_O_3_ particles for MAO 30 min. In order to avoid the interference of Al elements contained in Ti6Al4V, pure titanium (TA2) was used as the substrate material.

Figure 8a shows the cross-sectional morphologies of the obtained composition coatings. ZrO_2_ and γ-Al_2_O_3_ particles were not found in the MAO coating, and no obvious interface was found between the two-step MAO process. Figure 8b–e show the distribution of Ti, O, Al, and Zr elements on the MAO coating, respectively. Al and Zr elements come from ZrO_2_ and γ-Al_2_O_3_, respectively. The particle sizes of ZrO_2_ and γ-Al_2_O_3_ are 2.21 µm and 0.05 µm, respectively, but there is no enrichment of elements in the picture. Furthermore, the Zr element and the Al element did not delaminate during the two-step MAO process, which proves that the composite coating was not grown in layers, but was repeatedly melted and solidified in a way that penetrated the MAO coating. Each breakdown (high temperature melting) and solidification of the molten pool is a reorganization that penetrates the coating. The ZrO_2_ and γ-Al_2_O_3_ particles dissolved under the high temperature of the molten pool, which indicates that the micro-arc oxidation composite coating strengthening mechanism is elemental doping, rather than particle enhancement.

Figure 9 shows a typical SEM image of the MAO coating. Due to the temperature field of the molten pool, a crater-like morphology is formed. The temperature field gradient of the molten pool is large, and the highest temperature is in the middle of the MAO coating. The surface of the MAO coating remains solid due to the strong cooling effect of the electrolyte. Molten oxide is sprayed from the middle of the coating to the surface, and after freezing, the morphology is formed, as shown in Figure 9. The melt quickly evaporates the water from the surrounding electrolyte during the cooling process, so that the secondary phase particle is enriched and adhered to the surface, and is sandwiched into the MAO coating during the next period of molten pool formation.

## 4. Conclusions

The secondary phase particles in the MAO composite coating generate a phase transformation under the effect of the molten pool temperature field in the Ti6Al4V titanium alloy. The reason for the multiphase coexistence is that the high-temperature phase is retained by the rapid cooling effect of the electrolyte, and the low-temperature phase is formed due to the second phase transformation during the long active time of the molten pool temperature field.

For the temperature field distribution range of the molten pool in the active period, the lower limit is 2123 K and the upper limit is no lower than 3683 K. The continuous active time of the molten pool is shorter in the initial stage of the MAO process, which increases with the increase of the coating thickness. At the same time, with the extension of the MAO time, the size of the electric spark and the molten pool also becomes larger. The temperature field gradient of the molten pool is huge.

The strengthening mechanism of the composite coating prepared by adding secondary phase particles is elemental doping, rather than particle enhancement. The way in which the secondary phase particles enter the composite coating is by adhering to the surface during the cooling process of the molten pool. In the next active period of the molten pool, the secondary phase particles are then wrapped into the coating.

## Figures and Tables

**Figure 1 materials-13-00464-f001:**
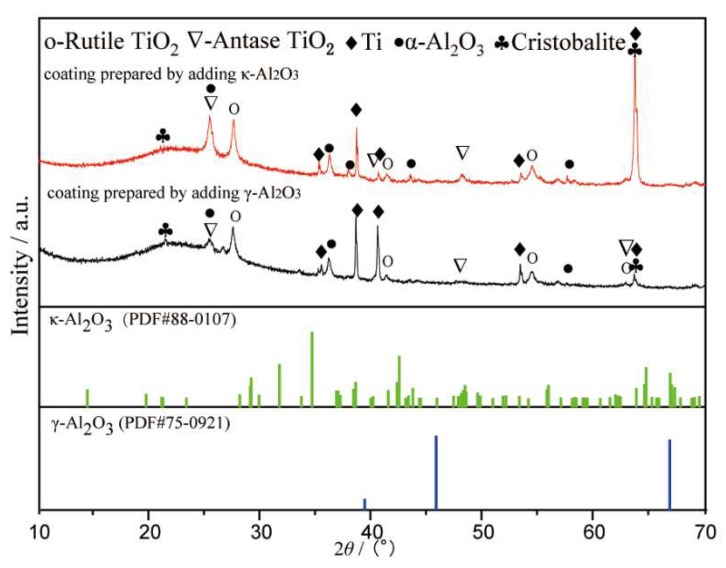
XRD pattern of the composite coatings prepared by adding γ-Al_2_O_3_ and κ-Al_2_O_3_.

**Figure 2 materials-13-00464-f002:**
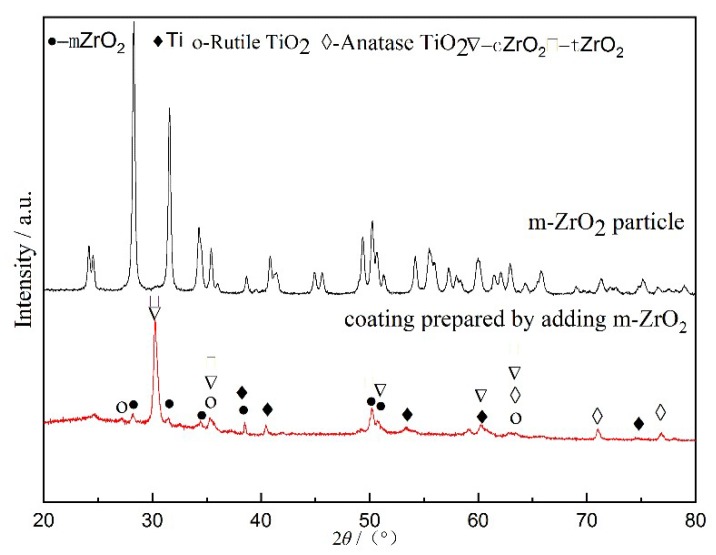
XRD pattern of the composite coating prepared by adding m-ZrO_2_.

**Figure 3 materials-13-00464-f003:**
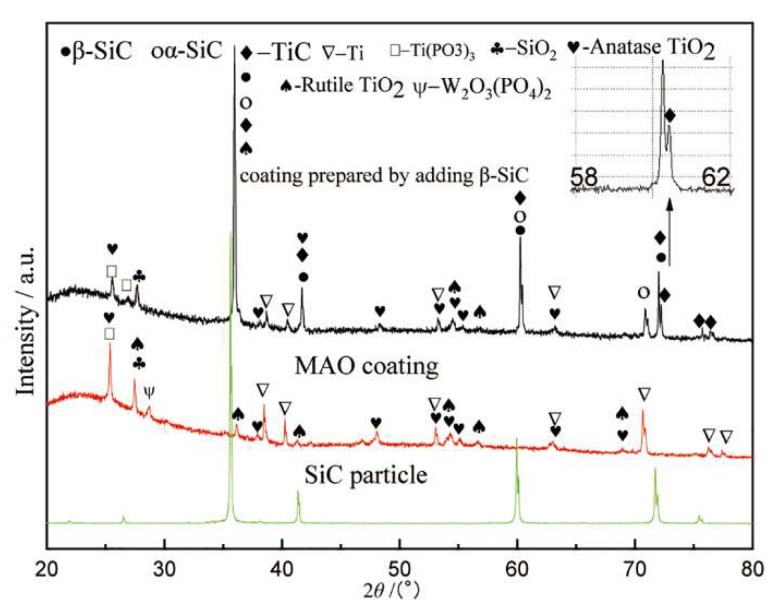
XRD pattern of the composite coating prepared by adding β-SiC.

**Figure 4 materials-13-00464-f004:**
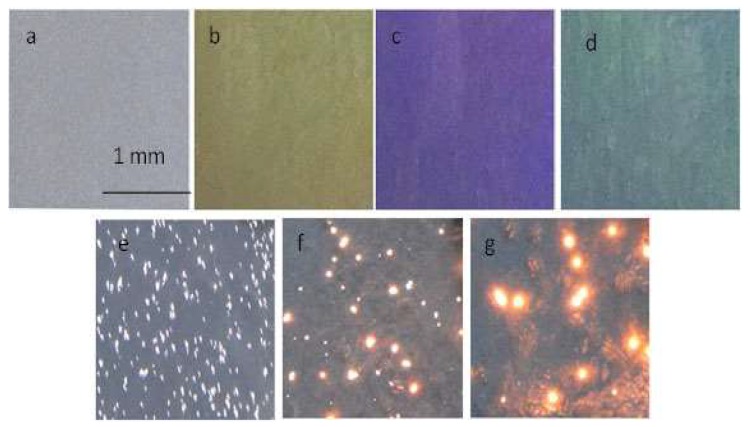
Ti6Al4V spark morphology of the micro-arc oxidation (MAO) process: (**a**) Ti6Al4V substrate; (**b**–**d**) anodizing; (**e**) 5 min; (**f**) 20 min; (**g**) 65 min.

**Figure 5 materials-13-00464-f005:**
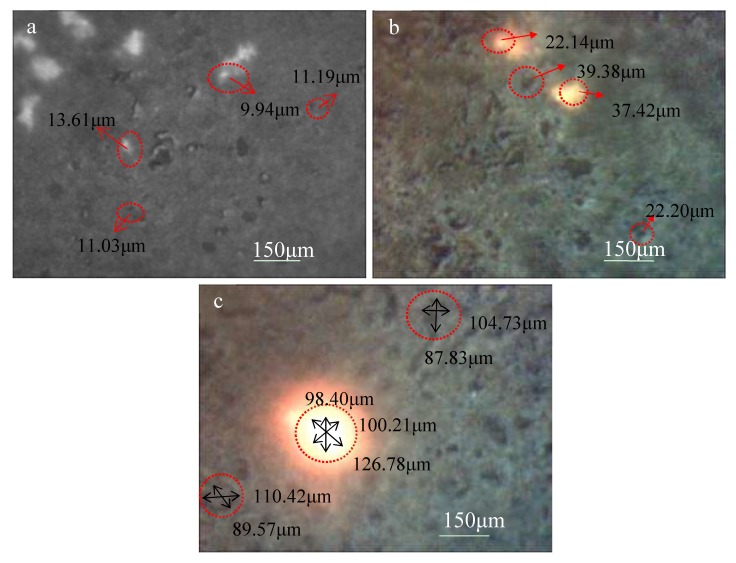
Optical photograph of the spark and the molten pool at typical moments: (**a**) 5 min; (**b**) 20 min; (**c**) 65 min.

**Figure 6 materials-13-00464-f006:**
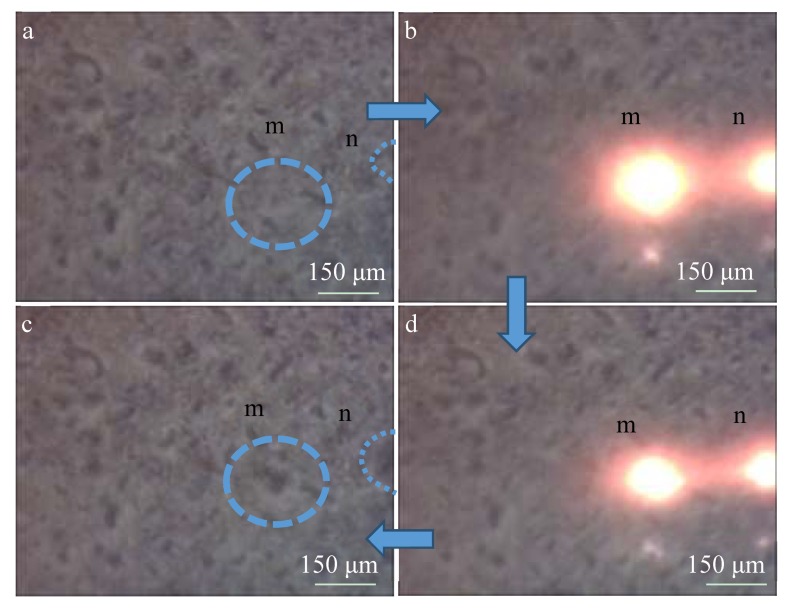
Surface images of the MAO coating before and after the spark discharge at 50 min: (**a**) 0 s; (**b**) 0.14 s; (**c**) 0.91 s; (**d**) 0.84 s.

**Figure 7 materials-13-00464-f007:**
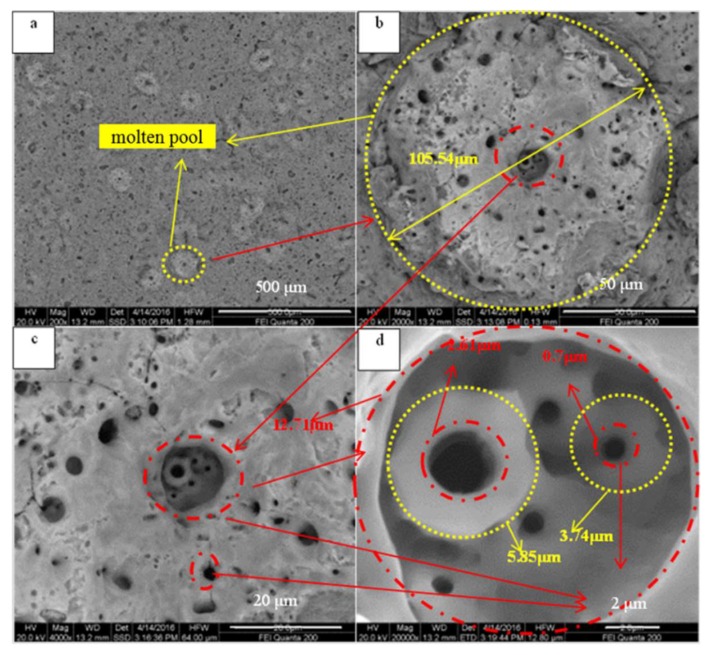
SEM image of the molten pool with details: (**a**) 200×; (**b**) 2000×; (**c**) 4000×; (**d**) 20,000×.

**Figure 8 materials-13-00464-f008:**
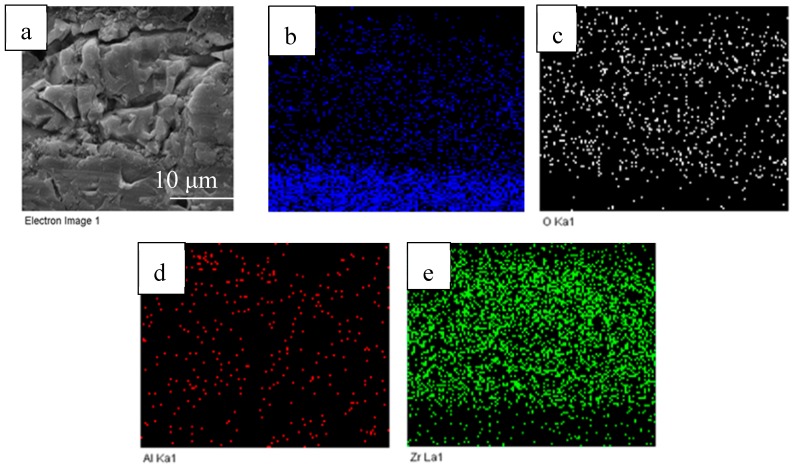
Cross-sectional SEM photograph and EDS distribution of the composition coating: (**a**) SEM image; (**b**) Ti; (**c**) O; (**d**) Al; (**e**) Zr.

**Figure 9 materials-13-00464-f009:**
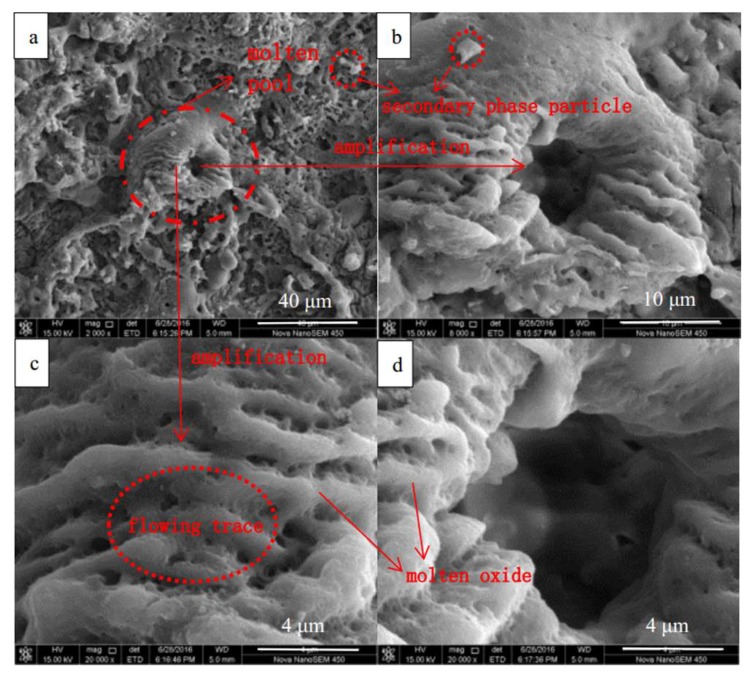
Surface morphology of the MAO coating.

**Table 1 materials-13-00464-t001:** Chemical composition of the Ti6Al4V alloy.

Type	Composition/wt.%					
	Al	V	Fe	C	O	Ti
**TC4**	**5.5–6.8**	**3.5–4.5**	**0.3**	**0.01**	**0.03**	**balance**

**Table 2 materials-13-00464-t002:** The characteristics of secondary phase particles.

Secondary Phase Particle Type	Average Size (µm)	Phase Transition Temperature	Phase Transition Product
γ-Al_2_O_3_	0.05	1303 K	α-Al_2_O_3_
κ-Al_2_O_3_	7	1288 K	α-Al_2_O_3_
m-ZrO_2_	2.21	1473 K/2643 K [22]	t-ZrO_2_/c-ZrO_2_
β-SiC	7	2373 K/3143 K	α-SiC/decompose

**Table 3 materials-13-00464-t003:** Electrolyte composition.

Composition	Concentration (g/L)
Na_2_SiO_3_·9H_2_O	8
(NaPO_3_)_6_	6
Na_2_WO_4_·2H_2_O	4
Na_5_P_3_O_10_	3
Secondary phase particles	6

**Table 4 materials-13-00464-t004:** The characteristics of the spark and molten pool at typical moments.

The Moments of MAO (min)	The Duration of the Spark (s)	The Density of the Spark (number/cm^2^)	The Size of the Spark (μm)	The Size of the Molten Pool (μm)
5	0.07	132	9.90	12.65
20	055	18	23.89	28.84
65	3.32	4	108.46	112.56

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
