# Peer review of "Characteristics of the Molten Pool Temperature Field and Its Influence on the Preparation of a Composite Coating on a Ti6Al4V Alloy in the Micro-Arc Oxidation Process"

_materials, 2020, doi:10.3390/ma13020464_

Round 1
Reviewer 1 Report
This is every interesting paper, where the MAO was applied to form the oxide layer with incorporated ceramic particles. The spark dischargers influence on the phase transformation and final phase composition of the coating.
However, some points in the paper should be improved and explained.
Aim of the work should be better presented and the introduction should better introduce into the scientific problem XRD pattern should be better described, especially that in Fig. 1 we observed the amorphous phase which increased after the MAO process, but this was not comment in the paper. Did you analyze the coating using another technique? Some of the peaks from k-Al2O3 do not correspond with the peaks presented for the k-Al2O3 coatings, why ? Why did you conclude that B-SiC was decomposed into Si and C ? There is no explanation when we analyze the XRD pattern, please clarify.
Author Response
Response to Reviewer 1 Comments
Thank you very much for your suggestion on our paper. We greatly appreciate the valuable comments by the reviewers. We have carefully read through the revision opinion of reviewers and improved our manuscript. All changes in the revised manuscript have been shown in Blue. The supplements according to the specific comments are highlighted in the revised manuscript.
Itemized response about reviewers’ comments:
This is every interesting paper, where the MAO was applied to form the oxide layer with incorporated ceramic particles. The spark dischargers influence on the phase transformation and final phase composition of the coating.
However, some points in the paper should be improved and explained.
Point 1: Aim of the work should be better presented and the introduction should better introduce into the scientific problem.
Response 1: Thanks for the reviewer’s kind advice. This work aimed to study the characteristics of the molten pool and its impact on the coating preparation in the micro-arc oxidation process. I have refined the introduction of the manuscript in the latest version.
Point 2: XRD pattern should be better described, especially that in Fig. 1 we observed the amorphous phase which increased after the MAO process, but this was not comment in the paper.
Response 2: Thanks for your careful review and constructive suggestion. In previous versions, the interpretation of XRD patterns was not clear enough. Figure 1 shows the XRD patterns of the coatings prepared by adding κ-Al2O3 and γ-Al2O3 particles. Neither κ-Al2O3 nor γ-Al2O3 exist in the as-prepared coatings, indicating that they were all transformed into α-Al2O3. The amorphous phase is formed during the MAO process under the effect of discharge with high energy.
Point 3: Did you analyze the coating using another technique? Some of the peaks from k-Al2O3 do not correspond with the peaks presented for the k-Al2O3 coatings, why ?
Response 3: Thanks for your careful reviews. I did not analyze the coating using another technique. In the latest version of Figure 1, PDF card 88-0107 representing κ-Al2O3 was used to analyze the composition of the coating prepared by adding κ-Al2O3. The results show that there is no κ-Al2O3 in the as-prepared coating, indicating that all of the κ-Al2O3 was transformed into α-Al2O3.
Point 4: Why did you conclude that B-SiC was decomposed into Si and C ? There is no explanation when we analyze the XRD pattern, please clarify.
Response 4: As shown in Figure 3, the as-prepared coating included β-SiC, α-SiC, and TiC. According to Table 2, β-SiC will be transformed into α-SiC at 2373 K, and will decompose into Si and C at a temperature higher than 3143 K. The existence of TiC illustrates that β-SiC has been decomposed into Si and C.
Now we have revised and polished the language of this article. Moreover, the English throughout the manuscript has been checked and improved by MDPI. All changes in the revised manuscript have been shown in BLUE.

Reviewer 2 Report
This work deals with incorporation of external particles into the MAO coating composition as a temperature markers. However, authors did not pay enough attention to the fact that phase transformations in course of MAO do not directly correlate with the melting-solidification processes in the bulk materials. The main reason for this is presence of the high electric field conditions that cause breakdown and internal stresses in the film. The fact that high temperature phases can be obtained at much smaller temperature (than bulk transformation requires) in MAO illustrates significant effect of electric field on the phase transitions in the coating under MAO conditions. Moreover, following questions and remarks should be considered before publication.
Please put substrate material into title and abstract. Title and abstract mentioned existence of “molten pool”, is any evidence of liquid phase? It is unclear what “second phase” does mean. What is the first phase? Usually, MAO coatings are multiphase even without any additional external particles. It includes amorphous and various crystalline phases (rutile, anatase). I would suggest replace “second phase” with “secondary phase”. Moreover, this term must be properly introduced and explained. The set of particles was not explained. Why did you use those particles? K-Al2O3? The reason for mixing the particles with NaOH and anionic surfactant is unclear. Process duration was not pointed. Table 4 is unclear. Why were crystal structures characterised in different way in respect to table 2? Why was some product characterised by formula, some by geometry? Equations 1 and 2 duplicate data from table 4. 5. I cannot see any molten pool on the figure. I can see sparks and pores, but how had authors identified a molten pool? Later, authors site ref [25] to support existence of molten pool, but that ref does not contain any evidence, it includes only speculative discussion without any confirmations of molten states. And fig 7 does not show something molten. Such morphology could be also produced by solidification of oversaturated liquid. Cross sectional images for Al2O3 and SiC particles were not presented. “...long active time of the molten pool temperature field.” it is unclear. What is “long time of the field”? Ref[1] has wrong spelling. Correct is “Malyschev V.N. Mikrolichtbogen-Oxidation - Ein neuartiges Verfahren zur Verfestigung von Aluminiumoberflaechen. // Metalloberflaeche, 1995, â„– 8, S.606-608.”
Author Response
Thank you very much for your suggestion on our paper. We greatly appreciate the valuable comments by the reviewers. We have carefully read through the revision opinion of reviewers and improved our manuscript. All changes in the revised manuscript have been shown in Blue. The supplements according to the specific comments are highlighted in the revised manuscript.
Itemized response about reviewers’ comments:
This work deals with incorporation of external particles into the MAO coating composition as a temperature markers. However, authors did not pay enough attention to the fact that phase transformations in course of MAO do not directly correlate with the melting-solidification processes in the bulk materials. The main reason for this is presence of the high electric field conditions that cause breakdown and internal stresses in the film. The fact that high temperature phases can be obtained at much smaller temperature (than bulk transformation requires) in MAO illustrates significant effect of electric field on the phase transitions in the coating under MAO conditions. Moreover, following questions and remarks should be considered before publication.
Point 1: Please put substrate material into title and abstract.
Response 1: Thanks for the reviewer’s kind advice. I have put the substrate material Ti6Al4V alloy into the title and abstract.
Point 2: Title and abstract mentioned existence of “molten pool”, is any evidence of liquid phase?
Response 2: Thanks for the reviewer’s question. During the MAO process, the specimen is immersed in a certain solution, and a discrete spark discharge occurs on the surface when the supplied voltage is higher than a critical value as a result of a loss in its dielectric stability in a region of low conductivity. The melted materials were ejected out of the channel at high discharge temperatures and pressures and cooled rapidly due to their immediate contact with the electrolyte, leading to an overall increase in the coating thickness [1,2]. Therefore, the molten pool becomes a liquid state in an extremely short time. During our observations, we noted that the molten pool had become a solid state with a crater-like morphology due to the cooling effect of the electrolyte.
O. Snizhko, A.L. Yerokhin, A. Pilkington, et al. Anodic processes in plasma electrolytic oxidation of aluminium in alkaline solutions. Electrochim. Acta 2004, 49(13): 2085-2095. H. Wang, M.H. F.Z. Du, et al. Effects of the ratio of anodic and cathodic currents on the characteristics of micro-arc oxidation ceramic coatings on Al alloys. Appl. Surf. Sci. 2014, 292: 658-664.
Point 3: It is unclear what “second phase” does mean. What is the first phase? Usually, MAO coatings are multiphase even without any additional external particles. It includes amorphous and various crystalline phases (rutile, anatase). I would suggest replace “second phase” with “secondary phase”. Moreover, this term must be properly introduced and explained.
Response 3: Thanks for your careful review and constructive suggestion. I have replaced the term “second phase” with “secondary phase” and explained this term in the latest version.
Point 4: The set of particles was not explained. Why did you use those particles? K-Al2O3? The reason for mixing the particles with NaOH and anionic surfactant is unclear. Process duration was not pointed.
Response 4: Based on the purpose of this paper, the range of the molten pool temperature field was estimated by the phase transition temperature of different secondary phases. The reason for choosing the set of particles is that Al2O3, ZrO2, and SiC are commonly used as secondary phase particles to improve the properties of coatings. κ-Al2O3 has been used in an astounding array of technological applications due to its good thermal, chemical, dielectric, and wear-resistant properties [3]. The reason why we chose κ-Al2O3 is that we wanted to obtain those good properties. The role of NaOH is to degreasing. Additionally, the role of the anionic surfactant is to disperse the particles.
Tseng W J , Chang J H . Preparation of κ-Al2O3/resin composite coating on polyethylene terephthalate foil for gas-permeation barrier[J]. Ceramics International, 2014, 40(10):16779-16784.
Point 5: Table 4 is unclear. Why were crystal structures characterised in different way in respect to table 2? Why was some product characterised by formula, some by geometry? Equations 1 and 2 duplicate data from table 4. 5.
Response 5: The previous Table 2 and 4 did not indicate this clearly, so I combined and integrated the two data. The results are as follows.
Table 2. The characteristics of secondary phase particles.
|
Secondary phase particle type |
Average size (µm) |
Phase transition temperature |
Phase transition product |
|
γ-Al2O3 |
0.05 |
1303 K |
α-Al2O3 |
|
κ-Al2O3 |
7 |
1288 K |
α-Al2O3 |
|
m-ZrO2 |
2.21 |
1473 K /2643 K [23] |
t-ZrO2/ c-ZrO2 |
|
β-SiC |
7 |
2373 K /3143 K |
α-SiC/ decompose |
Point 6: I cannot see any molten pool on the figure. I can see sparks and pores, but how had authors identified a molten pool? Later, authors site ref [25] to support existence of molten pool, but that ref does not contain any evidence, it includes only speculative discussion without any confirmations of molten states. And fig 7 does not show something molten. Such morphology could be also produced by solidification of oversaturated liquid.
Response 6: Holes were generated by the discharge channel, and the melted materials ejected out of the discharge channel to meet the electrolyte solidification. The molten pool had a crater-like morphology. Because the melted material was inside the coating, once it came out and made contact with the electrolyte, it became solid. Therefore, we could not directly observe the molten state. The molten pool exists in a liquid state in an extremely short time. If the molten state needs to be captured, further research is needed.
Point 7: Cross sectional images for Al2O3 and SiC particles were not presented. “...long active time of the molten pool temperature field.” it is unclear. What is “long time of the field”?
Response 7: This point is explained in the original manuscript. The size of the particles we chose is shown in Table 2. The secondary particles become a molten state under the effect of a high temperature. After repeated MAO process, the various components of the coating are very uniform. Therefore,cross-sectional images for Al2O3 and SiC particles were not presented. As shown in Table 4 in the latest version of the manuscript, the discharge time of a spark is 3s at 65 min. Compared with the initial period, the action time is longer.
Point 8: Ref [1] has wrong spelling. Correct is “Malyschev V.N. Mikrolichtbogen-Oxidation - Ein neuartiges Verfahren zur Verfestigung von Aluminiumoberflaechen. // Metalloberflaeche, 1995, â„– 8, S.606-608.”
Response 8: Thanks for your careful review. I have corrected it in the latest version.
Now we have revised and polished the language of this article. Moreover, the English throughout the manuscript has been checked and improved by MDPI. All changes in the revised manuscript have been shown in BLUE.

Round 2
Reviewer 1 Report
Authors made corrections, and the manuscript much better presents the XRD results and discussion about the MAO process. The process and the results are clear, and the manuscript is acceptable for publication.
Author Response
Dear Reviewer,
Thank you very much for your careful review of our manuscript. We greatly appreciate the valuable comments which greatly improved our paper.
Best regards
Qing Zhao

Reviewer 2 Report
Now manuscript sounds satisfactory. The only thing is that process duration should be clearly indicated in the experimental section.
Author Response
Dear Reviewer,
Thank you very much for your careful review of our manuscript. We greatly appreciate the valuable comments which greatly improved our paper. The changes in the first revised manuscript have been shown in Blue, and the changes in the second revised manuscript have been shown in Red. The supplements according to the specific comments are highlighted in the revised manuscript.
Itemized response about reviewers’ comments:
Point 1: Now manuscript sounds satisfactory. The only thing is that process duration should be clearly indicated in the experimental section.
Response 1: Thanks for your careful review and constructive suggestion. The process duration was supplemented in the second revised manuscript and presented in Red.
(-p3 the first paragraph: “The MAO time was 65 min. The choice of MAO time was based on the aim to study the duration and distribution of the spark and molten pool.”)
Best regards
Qing Zhao
